# Latitude-dependent oxygen fugacity in arc magmas

Fangyang Hu [1]✉, Hehe Jiang[1], Bo Wan [1], Mihai N. Ducea[2,3], Lei Gao [4] & Fu-Yuan Wu [1,5]

The redox state of arc mantle has been considered to be more oxidized and diverse than that of the mid-ocean ridge, but the cause of the variation is debated. We examine the redox state of the Cenozoic global arc mantle by compiling measured/calculated $fO_2$ of olivine-hosted melt inclusions from arc magma and modeled $fO_2$ based on V/Sc and Cu/Zr ratios of arc basaltic rocks. The results indicate that the redox state of Cenozoic arc mantle is latitude dependent, with less oxidized arc mantle in the low latitudes, contrasting with a near constant across-latitude trend in the mid-ocean ridges. We propose that such a latitude-dependent pattern in the arc mantle may be controlled by the variation in the redox state of subducted sediment, possibly related to a latitudinal variation in the primary production of phytoplankton, which results in more organic carbon and sulfide deposited on the low-latitude ocean floor. Our findings provide evidence for the impact of the surface environment on Earth's upper mantle.

Continental crust forms in volcanic arcs, and its compositional evolution is closely related to the oxygen fugacity ($fO_2$) of arc magmas[1–3]. The $fO_2$ of arc mantle has been believed to be higher than that of the mid-ocean ridge in general, possibly due to the subduction of more oxidized surface material into the deep Earth[1,4,5]. However, according to several global compilations, the redox state of the Cenozoic arc mantle is highly variable, and the cause of the variation is still under debate[1,4,6]. Three mechanisms have been proposed. The first suggests that the mass transfer of oxidant (e.g., sulfate) from the slab to the mantle is the primary cause of oxidation of the arc mantle, which is independent of slab thermal structure[7,8]. The second suggests that the efficiency of the slab-derived fluid (oxidant) added to the mantle wedge depends on the subducted slab's thermal structure. For example, old subducted slabs with steep dips could facilitate the release of more oxidant to the mantle wedge due to elevated fluid or sulfur fluxes[6,9]. The third suggests that subducted sedimentary rock is a redox filter for slab-derived fluids[10], and the redox state of sedimentary rock may modulate the

oxygen fugacity of the mantle wedge[11]. Regardless of the cause, the implication from these models is that the $fO_2$ diversity of subducted slab may play an important role in controlling the global $fO_2$ variation in arc mantle.

Of particular note is that the major difference in the composition of subducted slab lies in the subducted sediments[11,12]. As seafloor deposition is closely linked to oceanic biological productivity, the composition of oceanic sedimentary rocks could be latitude dependent. Phytoplankton are the major source of oxygen and organic carbon in the ocean[13]. The distribution of its species richness[14] and the associated primary production[15], as indicated by the sea surface temperature (SST), show a symmetric pattern along latitudes (Fig. 1). Such a pattern may influence the amount of organic carbon, a major reductant, in oceanic sediments, which may help reduce the sulfate in the oceanic crust[16]. However, the burial flux of carbonate on the seafloor is higher in low-latitude regions, which is proposed as a major oxidant for the sub-arc mantle[17] (Fig. 1). These latitudinal variations in the distribution of phytoplankton and carbonate may regulate both

[1]State Key Laboratory of Lithospheric and Environmental Coevolution, Institute of Geology and Geophysics, Chinese Academy of Sciences, Beijing, China. [2]Faculty of Geology and Geophysics, University of Bucharest, Bucharest, Romania. [3]Department of Geosciences, University of Arizona, Tucson, AZ, USA. [4]State Key Laboratory of Geological Processes and Mineral Resources, School of Earth Sciences and Resources, China University of Geosciences, Beijing, China. [5]College of Earth and Planetary Sciences, University of Chinese Academy of Sciences, Beijing, China. ✉e-mail: hufangyang@mail.iggcas.ac.cn

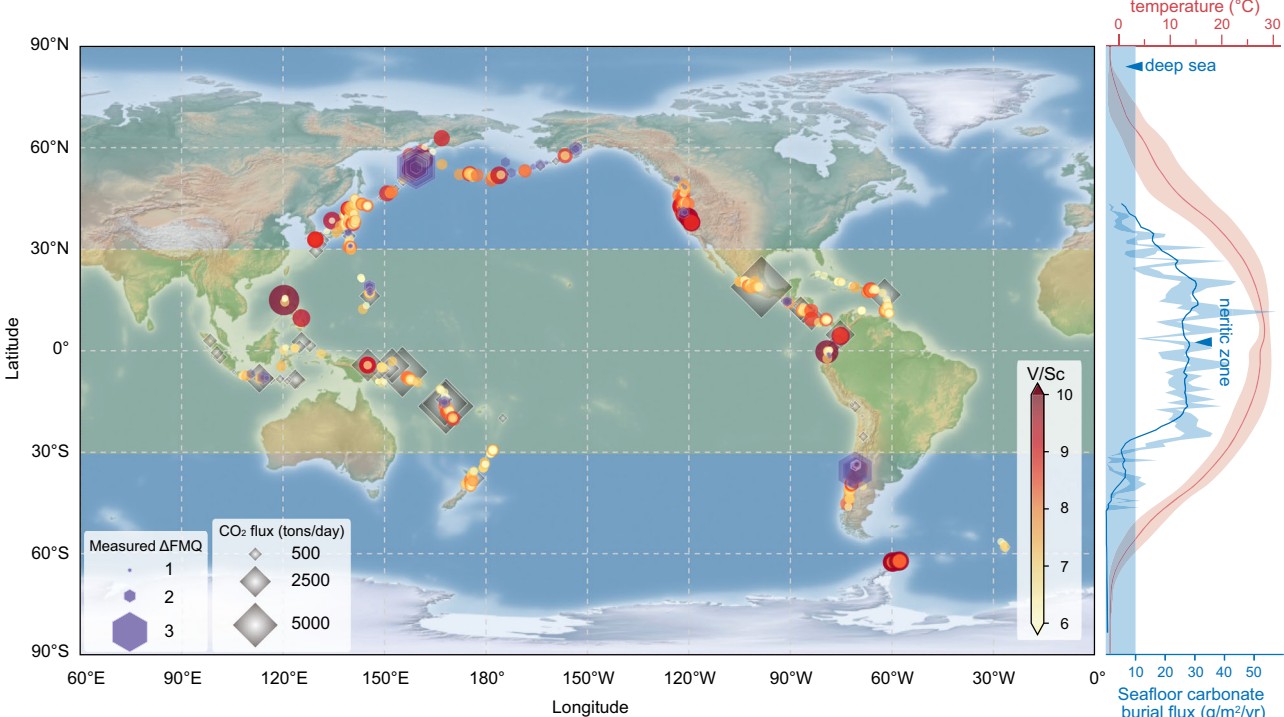

**Fig. 1 | Distribution of ΔFMQ of olivine-hosted melt inclusions and V/Sc ratios of arc basaltic rocks along the Rim of the Pacific.** The high ΔFMQ values and V/Sc ratios of arc basaltic rocks are preferentially distributed in high-latitude (>30 °N/S) regions (Supplementary Data 1 and 2). $CO_2$ flux emissions of volcanos are plotted on the map[72], showing that the giant emissions are mainly distributed in the low-latitude (<30° N/S) regions. The sea surface temperature data are the monthly average data from the NOAA, and the red shaded area represents uncertainties in 1 SD. The seafloor carbonate burial flux is plotted as a blue curve and shaded area based on data from O'Mara and Dunne[73]. The carbonate burial flux in deep sea is mainly lower than 10 g/m²/yr, whereas the flux in neritic zone decreases from low (>30 g/m²/yr) to high-latitude regions (<10 g/m²/yr) (Supplementary Data 3). The solid blue line represents the moving average results, and the blue shade represents the uncertainties of 1 SE.

the composition and redox state of oceanic sediments, affecting the arc mantle during subduction.

Here, we test the above hypotheses by evaluating the redox state of the Cenozoic arc mantle through the measured/calculated $fO_2$ of olivine-hosted melt inclusions as well as two independent oxybarometers of V/Sc and Cu/Zr ratios of basaltic rocks on a global geochemical data set. The observed variation in measured/modeled $fO_2$ of arc basaltic rocks exhibits a strong latitudinal pattern (Fig. 2, Supplementary Fig. 1), supporting that the redox state of the arc mantle could be regulated by a latitude-dependent mechanism driven by marine organisms. We propose that the arc evolution results from the interplay between the deep Earth and surface environment.

## Results

### Redox state of olivine-hosted melt inclusions in arc basalts
The redox state of basaltic rocks can be represented by the $fO_2$ of olivine-hosted melt inclusions, which can be obtained by measuring (1) the valence state of Fe and S of olivine-hosted melt inclusions, and (2) the V contents of olivine and olivine-hosted melt inclusions[3,18,19]. We compiled ~200 published data of measured/calculated ΔFMQ (log units relative to fayalite–magnetite–quartz buffer) of olivine-hosted melt inclusions from arc basaltic rocks (Supplementary Data 1). We grouped them based on the volcanoes/arc segments and methods of measurement. The calculated average ΔFMQ values of different arc segments by different methods all display a tendency to increase with latitude, ranging from ~+1.0 around the equator to ~+2.0 at high-latitude regions (Fig. 2a, b; Supplementary Fig. 1). Such a pattern indicates that the redox state of arc basalt could be latitude dependent. However, the melt inclusions in olivine could be captured at different times[20], such as crystal fractionation, magma mixing, and

degassing. Additionally, it has been argued that melt inclusions could be modified after entrapment[20,21]. Moreover, the available data from melt inclusions are relatively limited. As a result, additional proxies are needed to crosscheck whether the correlation between the redox state of arc basalt and latitude is significant.

### V/Sc and Cu/Zr as proxies of redox state
The V/Sc ratios of basaltic rocks have been considered to be relatively insensitive to low-temperature alteration, metamorphism, and the effects of differentiation and degassing for high-MgO primary basalts[1,22], and have been widely applied to reconstruct the $fO_2$ of magma sources[1,4,22–25]. V has variable valence states (2+, 3+, 4+, and 5+) and becomes more incompatible as its valence increases[4,24,25]. Sc is monovalent (3+), and its incompatibility does not change with the redox state[4,24,25]. Thus, the V/Sc ratio of basaltic rocks could record mantle $fO_2$ during partial melting, with a higher V/Sc ratio indicating a more oxidized arc mantle[1,4,24]. The thickened crust may lead to higher V/Sc ratios of arc magma than those in thin crust by leaving garnet as a residual mineral at the same $fO_2$. However, our compiled data show that the V, Sc, and V/Sc ratios have no correlations with Dy/Yb, La/Yb, and crustal thickness (Supplementary Fig. 2), reflecting that the effect of garnet on V/Sc ratios is neglectable. The V/Sc ratios of basalts are also sensitive to degrees of partial melting, partition coefficients of V and Sc, and initial V/Sc ratios of their mantle source[1,23], but if these parameters can be constrained, then V/Sc ratios of arc basalts are expected to be sensitive to variations in the $fO_2$ of a peridotitic source.

The Cu/Zr ratio has recently been proposed to be an indicator of the redox state of primary magma at given sulfur contents and melting degrees[26]. The sulfur solubility in basaltic magma increases substantially at higher $fO_2$, which leads to higher Cu contents[27]. Zr could

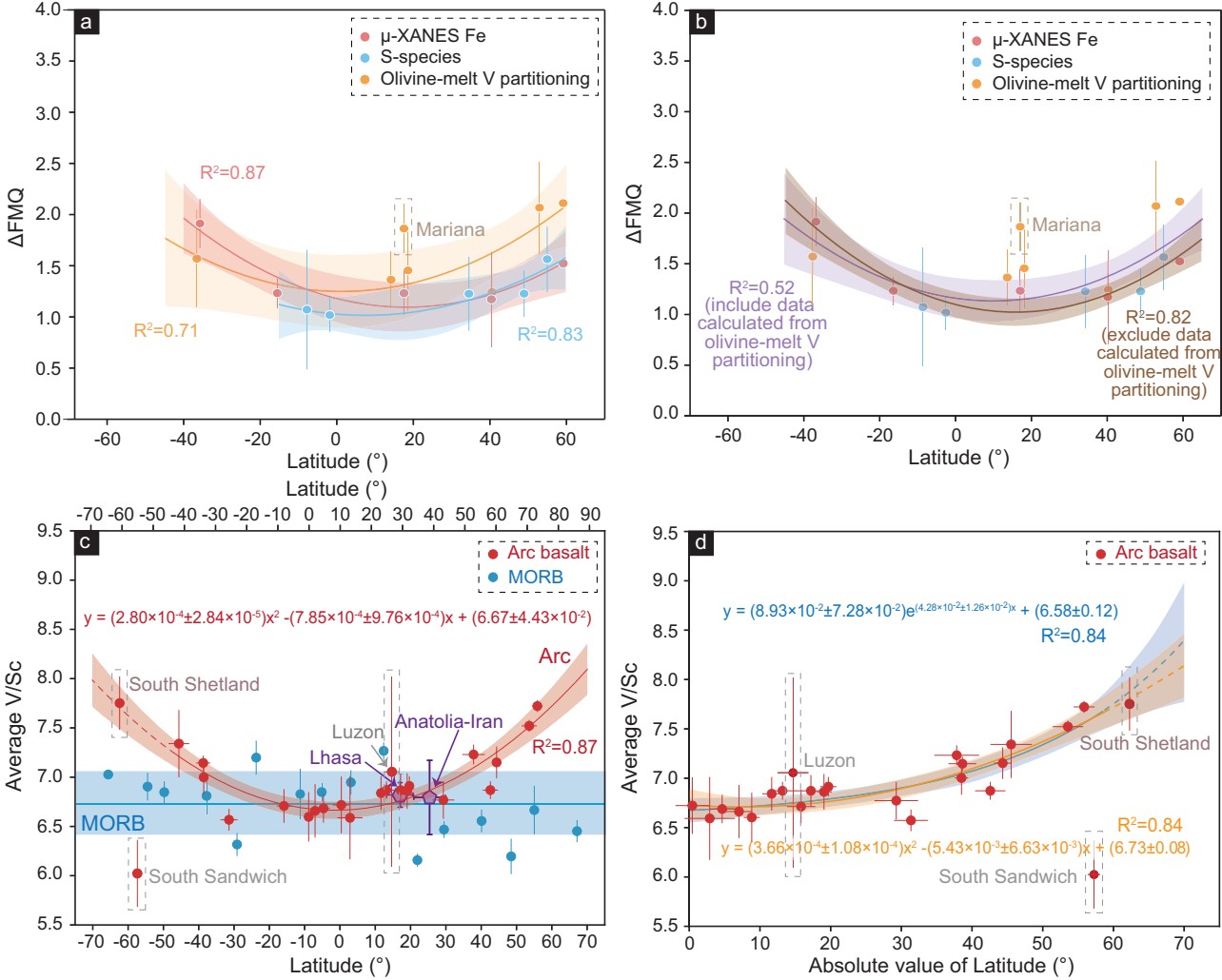

**Fig. 2 | Relationships between the ΔFMQ of olivine-hosted melt inclusions and V/Sc of basaltic rocks and latitude.** Plots of measured/calculated ΔFMQ of olivine-hosted melt inclusions by different methods versus latitude. The regressions were made for each of the different methods (**a**) and they have quadratic correlations with latitude ($R^2 > 0.70$). The regressions were also obtained for all data together (**b**), and the correlations were significant ($R^2 > 0.50$), confirmed by bootstrap Monte Carlo resampling results (Supplementary Fig. 3). The error bars of ΔFMQ and latitude are 1 SD. The data of Mariana arc from ref. 74 is excluded from the regressions. The compiled data are listed in Supplementary Data 1. **c** Plots of average V/Sc ratios of arc basaltic rocks and mid-ocean ridge basalts (MORB) versus latitude. A strong quadratic correlation ($R^2 = 0.84$) is obtained, confirmed by bootstrap Monte Carlo resampling results (Supplementary Fig. 3). The data of South Sandwich and Luzon

are excluded for regressions. The South Shetland data are included for regressions although their tectonic setting remains controversial. The V/Sc ratios of MORB remain constant across the latitude, and the shaded region represents the standard deviation of data. The Cretaceous basaltic rocks from the low-latitude Neo-Tethys (Lhasa terrane[75] and Anatolia-Iran belt[76]) are also displayed. The error bars of V/Sc ratios are 1 SE, and the error bars of latitude are 1 SD. The compiled global data are listed in Supplementary Data 2, 4, and 5. Source data are provided as a Source Data file. **d** Plots of average V/Sc ratios of arc basaltic rocks versus absolute values of latitude. A significant quadratic ($R^2 = 0.84$) or exponential relationship ($R^2 = 0.84$) is obtained. The shaded region of regressions in this figure represents a 95% confidence interval.

reflect the melting degree and is not influenced by the redox state of the source. The Cu/Zr systematics represent a chalcophile element melting behavior[26] that varies independently from the lithophile-dominant V/Sc systematics. Here, we use the Cu/Zr ratio as a secondary proxy to complement the V/Sc record. As previous work stated[26], despite being capable of implying $fO_2$ of the primary magma, the Cu/Zr ratio also correlates well with the Moho depth, reflecting a strong influence of melting fraction on this proxy (Supplementary Fig. 2). Therefore, the Cu/Zr ratio could not be as directly correlated to $fO_2$ as the V/Sc ratio[4,26].

We evaluate the variation in the V/Sc and Cu/Zr ratios using a forward modeling approach ("Methods"). With proper parameter set-ups (degree of melting, melting temperature–pressure, initial composition, modal mineralogy of the source, partition coefficients, and sulfur content at sulfide saturation (SCSS)[23,26], etc.), the inferred $fO_2$

values from the V/Sc and Cu/Zr ratios of the same basaltic rocks should converge.

## Latitudinal variation in the redox state of the Cenozoic arc mantle

We compiled the V/Sc and Cu/Zr ratios for the magmatic rocks from the Cenozoic circum-Pacific volcanic arcs and the mid-ocean ridges from the GEOROC database (Supplementary Data 2 and 4). We filtered the data for primary basaltic rocks to minimize the effects of crystal fractionation, crustal assimilation, magma mixing, and crystal cumulating (see "Methods"). We then calculated the mean values of each arc segment for the arc basaltic rocks or within each of the 10° latitude bins for the mid-ocean ridge basalts. The results are plotted in Fig. 2c, d. The mean V/Sc values of the primary arc basaltic rocks from different arcs display a strong quadratic or exponential relationship

with latitude ($R^2 = 0.84$): the V/Sc values are lowest around the equator, remain relatively constant at low latitudes (30° S to 30 °N), and then increase toward high latitudes (>50° N/S). In contrast, the mean V/Sc values of the primary MORB remain constant across all latitudes (Fig. 2c; Supplementary Data 4). The Cu/Zr ratios show similar contrasting behavior between the arcs and mid-ocean ridges (Supplementary Fig. 1), although the latitudinal dependency of arc basaltic rocks is not as strong as that displayed in the V/Sc ratios.

Variations in V/Sc ratios in basaltic rocks can be attributed to the following mechanisms, including: (1) oxidized weathering[28]; (2) source composition[23]; and (3) melting processes, in which the degree of partial melting[23,24] and the redox state of the mantle source[1,22] are the predominant controlling factors. Below, we evaluated each of them before extracting the oxygen fugacity information from the V/Sc and Cu/Zr ratios.

Sc is immobile during weathering, whereas V will be lost if weathering is under oxidized conditions[22,28]. Weathering in low-latitude regions is relatively more intense and may be more oxidized, which may influence the V/Sc ratio of basaltic rocks[29]. However, the data we collected here show relatively low loss-on-ignition (LOI) values (mostly <4 wt%) which are not latitude-dependent (Supplementary Fig. 1). Therefore, the latitudinal V/Sc trend in arc basaltic rocks presented here might not be a result of the difference in weathering intensity.

Arc basaltic rocks are more enriched in Sr–Nd isotopic composition than the MORB (Fig. 3a), which might be related to subducted

sediments[30,31] or upper plate contamination. Because our selected data represent primitive basaltic magma, the crustal contamination could be negligible. A mélange melting model suggests that the average addition of sediment in the subduction zone is ~1% (up to 10%, depending on the isotopic composition of sediments) (Fig. 3a; Supplementary Data 6), consistent with previous estimates[30,31]. Although the V, Sc, Cu and Zr contents of arc basaltic rocks do show weak correlations with the Nd isotopic compositions (Fig. 3c, d), there is no latitudinal variation in the Nd isotopic compositions of basaltic rocks in either arcs or mid-ocean ridges (Fig. 3b), indicating that the influence of source variations is insignificant. Therefore, the observed V/Sc trend in arc basaltic rocks (Fig. 2c) is more likely to result from melting processes.

The arc basaltic rocks display a wide range of melting degrees, likely due to the differences in the thermal structure of the subduction zone and (upper plate) lithospheric thickness[30,32]. The low-latitude arcs generally show higher melting degrees due to their comparatively thin crust (<30 km), resulting in higher Sc, V, and Cu contents and lower Zr and Ti contents than those in higher latitude arcs[26,32] (Supplementary Fig. 1). Such varied melting degrees make it difficult to infer $fO_2$ directly from the V/Sc and Cu/Zr ratios. Therefore, we modeled the partitioning of V, Sc, Ti, Cu, and Zr during mantle melting beneath the arc and the mid-ocean ridge at their typical ranges of mantle source composition, temperature, and pressure under different sets of melting degrees and redox conditions ("Methods"; Fig. 4; Supplementary Figs. 5 and 6).

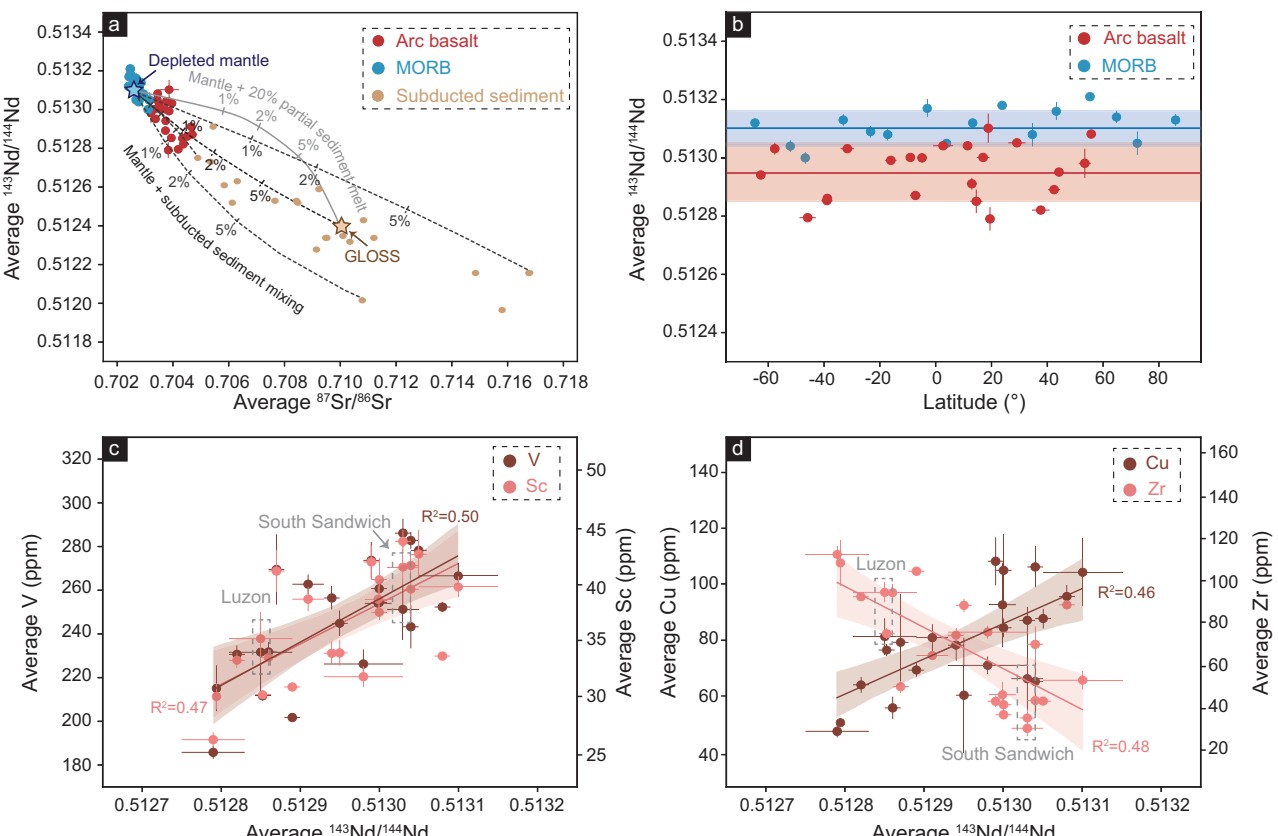

**Fig. 3 | Geochemical characteristics of compiled arc basaltic rocks and MORB.** The compiled data are shown in Supplementary Data 5. Source data are provided as a Source Data file. **a** Plot of average $^{143}Nd/^{144}Nd$ versus average $^{87}Sr/^{86}Sr$ of each arc segment and MORB. The arc basaltic rocks are more enriched than MORB, reflecting potential enrichment by subducted sediments[31,77]. Mélange melting model (black dotted lines) and mixing between mantle and 20% sediment melts (gray line) are shown for comparison. The modeling parameters are shown in Supplementary Data 6. **b** The variations of average $^{143}Nd/^{144}Nd$ of each arc segment and MORB across latitude. The thick lines and shaded ranges represent mean values and standard deviations. **c** Plot of average V (ppm) and Sc (ppm) versus average $^{143}Nd/^{144}Nd$ of each arc segment. The shaded region of regressions represents a 95% confidence interval. **d** Plot of average Cu (ppm) and Zr (ppm) versus average $^{143}Nd/^{144}Nd$ of each arc segment. The shaded region of regressions represents a 95% confidence interval. All error bars are 1 SE, except for error bars of latitude being 1 SD.

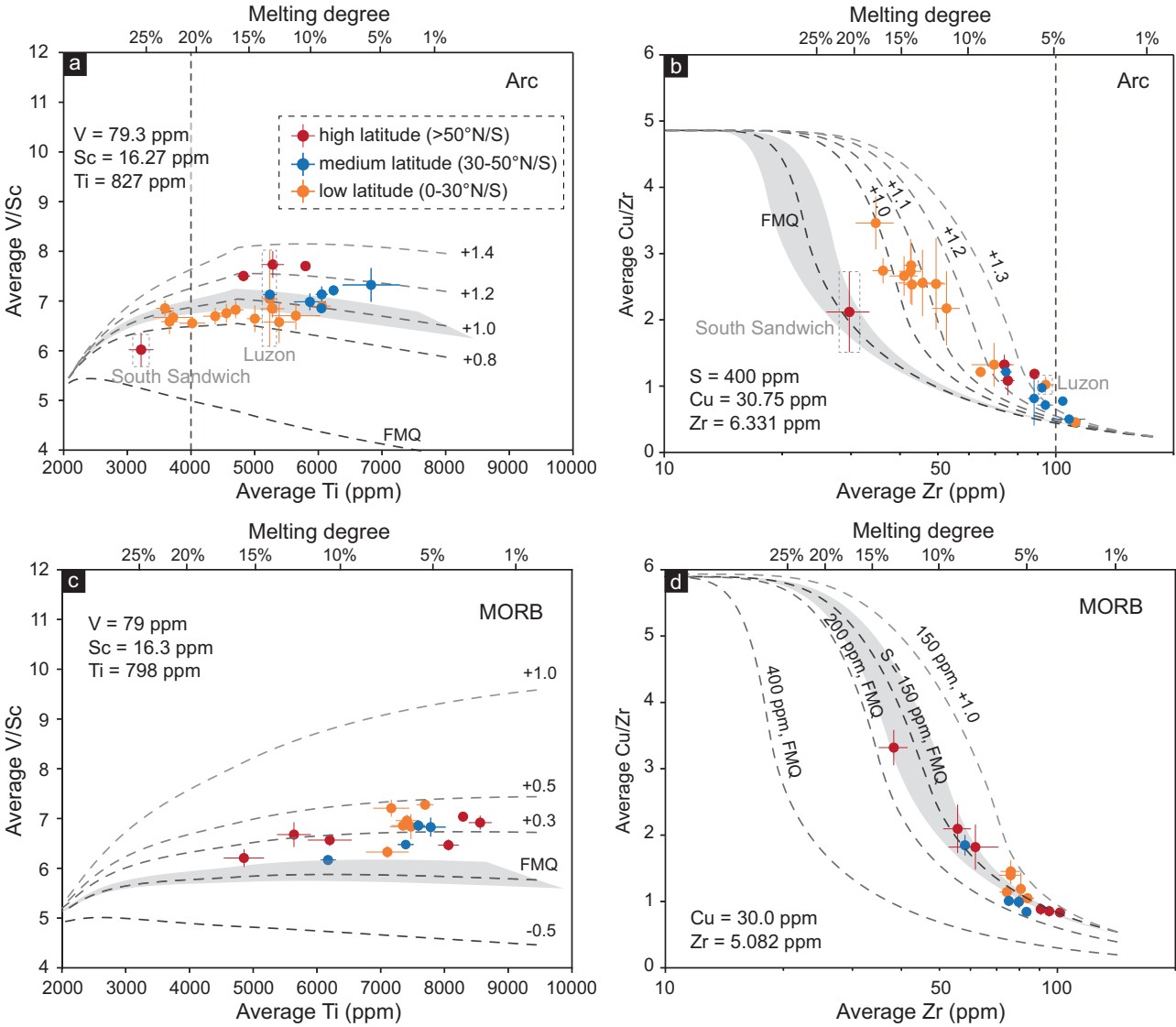

**Fig. 4 | Estimated oxygen fugacity of arc basaltic rocks and MORB.** The source composition of arc mantle is composed of 99% depleted mid-ocean ridge mantle[58,59] plus 1% average subducted sediments (GLOSS)[12]. **a, c** V/Sc versus Ti (ppm). The calculated $fO_2$ isopleths are modeled at a melting temperature of 1231 ± 33 °C and a melting pressure of 1.15 ± 0.24 GPa for arc basaltic rocks and a melting temperature of 1342 ± 29 °C and a melting pressure of 1.20 ± 0.20 GPa for MORB (Supplementary Fig. 4). The shaded regions represent the modeled V/Sc and Cu/Zr results under the estimated melting pressures and temperatures. The evaluation of source heterogeneity is shown in Supplementary Fig. 5. The melting degree is calculated from the average melting P–T conditions. **b, d** Cu/Zr versus Zr (ppm). The calculated $fO_2$ isopleths for arc basaltic rocks are modeled at $S$ = 400 ppm. The MORB data are modeled at $fO_2$ = FMQ with different mantle surfer contents ($S$ = 150 ppm, 200 ppm, and 400 ppm) and at ΔFMQ = +1.0 ($S$ = 150 ppm). The shaded regions represent 20% variations in sulfur solubility during mantle melting. The evaluation of source heterogeneity and sulfur content variations along latitude are shown in Supplementary Figs. 6 and 7. All error bars are 1 SE. The modeling results are shown in Supplementary Data 7–10. Source data are provided as a Source Data file.

As shown by the forward models, very small changes in V/Sc or Cu/Zr ratios may lead to quite different calculated ΔFMQ results when melting degree is high (>20%) for V/Sc ratios or low (<5%) for Cu/Zr ratios (Fig. 4; Supplementary Figs. 5 and 6). For example, the data from the Solomon arc, with low Ti content, were plotted on $fO_2$ isopleth with high $fO_2$ values in some of our model runs (Fig. 4; Supplementary Figs. 5 and 6). In fact, previous studies on mantle peridotite and primary basalts have shown that the Solomon arc mantle is relatively less oxidized[33] with an average ΔFMQ ≤ +1.0, which is consistent with our general observation that the arc basalt at low latitudes (<30° N/S) is less oxidized than that at higher latitudes. Therefore, we propose that the inferred ΔFMQ based on samples with Ti < 4000 ppm and Zr > 100 ppm may need caution (Fig. 4; Supplementary Figs. 5 and 6). The Cu/Zr-inferred $fO_2$ values of mid-latitude and high-latitude arcs are difficult to distinguish,

especially for those with high Zr contents. As a result, we focus on the results obtained by V/Sc ratios.

In Fig. 4, both proxies show that arc basalts fall within a modeled $fO_2$ range of ~+0.8 to 1.4 ΔFMQ, which is typical for arc mantle sources[4,5,26]. In contrast, data for the MORB plot in a narrower range of modeled melting extent and lower $fO_2$ values at ~0 to +0.3 ΔFMQ, consistent with previous studies[4,26], indicating that mid-ocean ridge mantle is less oxidized. In addition, both proxies show that arc basalt at higher latitudes (>30 °N/S) is more oxidized (ΔFMQ > +1.0 to +1.2) than that at lower latitudes (<30° N/S; ΔFMQ < +1.0 to +1.2) (Fig. 4). Such a pattern is not observed in the redox state of the MORB (Figs. 2 and 4). Therefore, the latitudinal variation in the implied redox state of arc mantle is significant.

The uncertainty in the $fO_2$ obtained through the above modeling approach includes (1) analytical uncertainties in the trace element

measurements, (2) uncertainties in the initial composition of mantle wedge, and (3) uncertainties in the partition coefficients. Collectively, the uncertainty of this method is -0.4–0.5 log units (1 SD) in ΔFMQ according to our evaluation (see Supplementary Information), consistent with previous estimations[34]. Among these, the partition coefficients, which are dependent on the empirical fitting coefficients in the functions, nonbridging O/tetragonal O ratio (NBO/T) of melt, chemical compositions of minerals, magma P–T conditions, oxygen fugacity[4], and initial compositions of mantle, are latitude independent. The change in these parameters will systematically modify the $fO_2$ isopleths, which will change the calculated absolute ΔFMQ values but have a limited effect on the ΔFMQ difference between the low-latitude and high-latitude arcs (Supplementary Fig. 5). Thus, the calculated differences in $fO_2$ between the high-latitude and low-latitude arcs could be identified, although the analytical error of trace elements could lead to -0.1–0.2 log unit variations in ΔFMQ.

## Discussion

It has been proposed that variations in slab thermal parameters (product of plate age, convergence velocity, and slab dip angle[35]) and/or the nature of fluids would result in variations in the redox state of the arc mantle[6,9,24,36]. The slab thermal parameters control the efficiency of slab dehydration[30,32], resulting in variations in the masses of slab-derived fluids fluxing into the mantle wedge[6,9], oxidizing it to variable extents. As such, correlations between the dehydration flux, slab thermal parameters, and the redox state of the arc mantle are expected (i.e., high $fO_2$ arc mantle in high-angle subduction zones). The Ba/Nb ratios of global arc magma have been used to trace dehydration processes in subducted slabs[37]. The results show that dehydration events occur predominantly at sub-arc mantle depths of -60, -120, and >200 km caused by the breakdown of various hydrous minerals[37]. Our compiled arc data show decreases in V/Sc and Ba/Nb ratios at slab dips from -10° to 20° and increases of V/Sc and Ba/Nb ratios at slab dips from -40° to 50° (Supplementary Fig. 8). In addition, increases in V/Sc and Ba/Nb ratios at -50 and -120 km mantle depths are observed (Supplementary Fig. 8). However, the overall trend for the V/Sc versus slab dip or depth to slab does not follow exactly with that in the Ba/Nb systematics, suggesting that the redox state of arc mantle is, at best, partially related to the slab thermal structure. In fact, the inferred and measured $fO_2$ for some modern arcs from previous studies suggest the presence of low $fO_2$ values in a steep subduction zone (e.g., parts of the Trans-Mexican arc)[6,38], and vice versa (e.g., parts of the Andes)[39]. There is also little correlation between slab thermal parameters and sulfur contents in the primary magma[7]. The above observations suggest that the slab thermal parameter might not be a direct cause for the latitudinal trend in the V/Sc ratios and inferred $fO_2$ of arc mantle in our study.

We propose that the latitudinal trend in the inferred redox state of arc mantle is more likely controlled by the variable intake of redox-sensitive elements in the slab-derived fluids, which is proposed to be regulated by the composition and redox state of the subducted sediments[10,11]. Recent experimental study by Tumiati et al.[40] suggests that the carbon isotope of volcanic gases is primarily controlled by the redox states of sediments and fluid-rock ratios, where less oxidized subducted sediments will lead to volcanic gases with higher $\delta^{13}C$ at a similar fluid-rock ratio, regardless of the organic/inorganic carbon ratio. Global compilation of modern gas emission from arc volcanoes suggests higher $\delta^{13}C$ at low latitude, supporting our hypothesis of less oxidized sediments at lower latitudes (Fig. 5a; Supplementary Fig. 9).

C, S, Fe, and Mn are the primary redox-sensitive elements contributing to the redox budget of the subducted sediment, which are mainly in the form of graphite/carbonate, sulfide/sulfate, and Fe–Mn nodules[7,11,41,42]. Among these elements, carbon and sulfur play the most important role, as the fluxes of their hypervalent state surpass other redox-sensitive elements significantly[17]. Therefore, it is important to

determine the influence of subduction of carbonate/organic carbon, and sulfide/sulfate on mantle redox state.

According to ref. 17, the mantle reference redox states of carbon and sulfur are $C^0$ and $S^{2-}$, respectively, and therefore the main contributions of carbon and sulfur to the mantle redox state are $C^{4+}$ and $S^-$. However, recent studies have shown that the carbon in sub-arc mantle could be stable as $C^0$ or $C^{4+}$[43]. Moreover, carbonate could remain stable during subduction[44]. As for sulfur, emerging research suggests that sulfate is a major oxidizing agent in subduction zones[7,45,46], although its global flux has yet to be properly quantified[17]. In addition, the dominant sulfide in marine sediments, namely pyrite, is also a prominent oxidizing agent for the sub-arc mantle[17]. We revised the redox budget model proposed by Evans[17] by adding the contribution fluxes of $C^0$ and $S^{6+}$, and reducing the contribution flux of $C^{4+}$ (see "Methods"). The revised redox budget shows that the calculated ΔFMQ varying between 0.3 and 1.2 log units at a time scale of 1–10 million years for subduction zones. This is broadly consistent with our observed and calculated ΔFMQ differences (0.4-1.0) for the sub-arc mantle (Figs. 2 and 4).

The marine burial of carbon is primarily controlled by the productivity of photosynthetic organisms[13], which displays strong latitudinal dependency[13,14] (Fig. 1). Significant latitudinal variation in the fluxes of particulate organic carbon accumulation is well observed[15] (Fig. 5b). If more organic carbon is subducted, then a less oxidized sub-arc mantle is expected. However, the content and flux of organic carbon in trench sediments do not show a similar distribution pattern to the POC (Fig. 5c, d). In comparison, the inorganic carbon, which is carbonate, is the predominate carbon in the trench sediments and its content and flux decrease from low-latitude regions to high-latitude regions[47] (Fig. 5c, d).

Current available datasets on subducted carbon do not appear to provide direct evidence on less-oxidized subducted sediments at low latitudes, but we propose that this conjecture is possible via sulfate reduction involving organic carbon[16]. Sulfur predominately serves as an oxidizing agent in the subduction process, no matter if it is sulfide ($S^-$) or sulfate ($S^{6+}$), or valences in-between. The main difference is that sulfate has a significantly higher capacity of oxidizing sub-arc mantle than sulfide. Limited sulfur isotope data of melt inclusions and basaltic rocks from subduction zones show that the primitive melts formed at lower latitude arcs have relatively lower $\delta^{34}S$ values, in comparison to consistent $\delta^{34}S$ values of MORB (Fig. 6). The $\delta^{34}S$ values of arc basaltic rocks are mainly controlled by the contributions from seawater sulfate and sedimentary sulfide[7,8]. Sulfate reduction involving organic carbon accompanied by the formation of sedimentary pyrite ($FeS_2$) is a common process in marine sediments[16], where enhanced POC deposition may lead to more sulfate reduced to sulfide at low latitudes. During this process, organic carbon will be consumed and transformed to carbonate[16], which may explain the discrepancy between the distribution patterns of POC in the ocean and organic carbon in trench sediments across latitude.

There are clearly limitations of our proposed mechanism for less-oxidized sub-arc mantle at the low latitudes; that to us suggests that the latitudinal control of oxygen fugacity of sub-arc mantle is controlled by more than one parameter/factor. First, the compiled data of POC, subducted sediments, sulfur and carbon isotope compositions of arc segments are mainly from limited sites, preventing us from obtaining a precise knowledge of their compositions globally. Second, the global distribution of $O_2$ beneath the ocean and the flux of carbon and sulfur in subduction zones are mainly model-based estimations. Therefore, there are apparent uncertainties in the calculated results. Third, although the hypotheses mentioned above are reasonable, the direct supporting data from global marine sediments or subducted sediments are still lacking, which clearly needs further studies to explore. Nevertheless, with all the caveats and limitations, our study shows that a less oxidized sub-arc mantle exists in the lower latitude

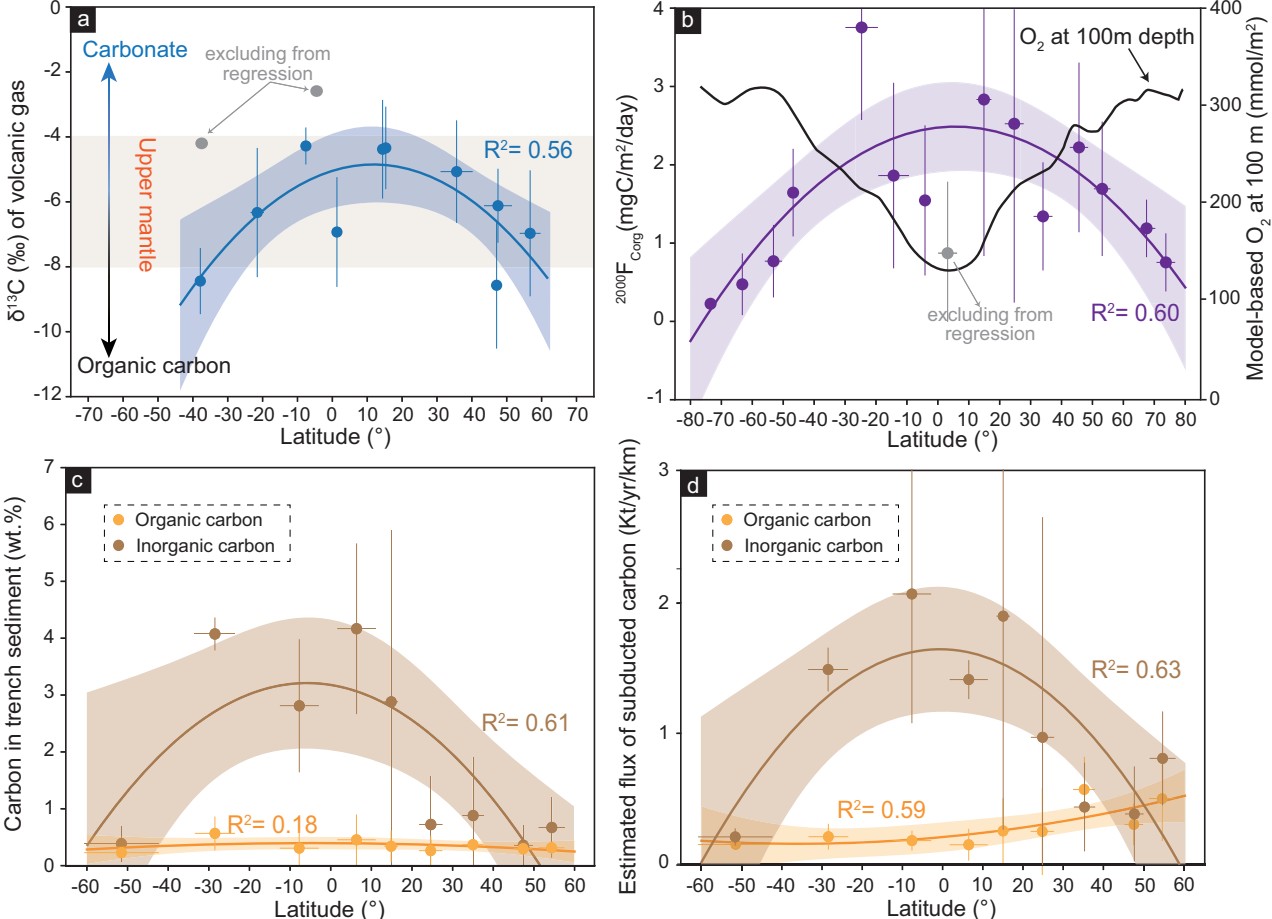

**Fig. 5 | δ¹³C of volcanic gases, flux of particulate organic carbon (POC) in the ocean, and carbon in subducted sediments. a** δ¹³C (‰) of volcanic gases across latitude. Original data are from ref. 78. The data with clear crustal assimilation signatures are excluded based on the carbon and helium isotope systems. The upper mantle range is based on ref. 78. The gray circles are excluded from regressions. **b** Average POC flux at 2000 m (purple circle) per 10° across latitude. The gray circles are excluded from regressions. Original data are from ref. 15. The

O₂ at 100 m depth (black line) is obtained from ref. 79. **c, d** Carbon in subducted sediment (wt%) and flux of subducted carbon (Kt/yr/km) across the latitude. Original data are from ref. 47 and plotted in Supplementary Fig. 10. The data were averaged with a bin of 10° or 20°. All shaded regions of regressions represent the 95% confidence intervals. All error bars are 1 SD. The compiled data are listed in Supplementary Data 12–14.

regions and is primarily related to the subducted sediments. No global tectonic parameters in the modern configuration would explain this. In combination of available data and known mechanisms, we propose that less oxidized subducted sediments in lower latitude regions could be related to the sulfate reduction caused by more organic carbon deposited on the lower latitude ocean floor. The subduction of more pyrite, which has lower oxidizing capacity than sulfate, may result in a less oxidized sub-arc mantle at lower latitude regions.

The mid-ocean ridge mantle has barely been modified by recycled materials[24], resulting in a constant redox state across latitudes. In contrast, in convergent margins where surface material is being transported into the mantle, the redox states in the arc mantle display systematic latitudinal variation, mimicking a typical temperature-driven signal often observed in the biosphere–hydrosphere. Although the fate of subducted sediments in subduction zones is still under debate, our findings suggest a strong link between biogenic redox-sensitive elements in the surface environment and the redox state of arc mantle, providing evidence on the impact of surface processes on Earth's interior processes, which might play an essential role in long-term mantle evolution (Fig. 7). In consideration of our findings and proposed model, subduction systems at different latitudes, e.g., Tethyan and Cordilleran orogenic systems[48,49], may exhibit contrasting resource and environment characteristics, which deserves further investigation.

## Methods
### Global data compilation
We complied the V/Sc and Cu/Zr ratios of arc basaltic rocks formed during the Cenozoic because the paleolatitudes of circum-Pacific arcs basically remained unchanged during the Cenozoic[50]. The whole-rock data of arc basaltic rocks are obtained from the GEOROC database, and the MORB data are from ref. 51. The V/Sc and Cu/Zr ratios of magmatic rocks will increase or decrease during magma differentiation processes, and therefore, it is crucial to obtain data of primary basaltic rocks. To ensure that the collected data could represent primitive melts, we only selected basaltic rocks (whole rocks and glasses) with $SiO_2 = 45–52$ wt%, $MgO = 6.5–15$ wt%, Mg# (molar $100 \times MgO/(MgO + FeO_T)$) = 60–72, and LOI < 6 wt% (Supplementary Fig. 11). According to our compilation, using such a criterion could minimize the effects of crystal fractionation, crustal assimilation, magma mixing, and crystal cumulating (Supplementary Figs. 12–14). In addition, to prevent interpreting oxygen fugacity complexity caused by compositionally diverse mantle sources, we excluded alkaline magmas (mostly basanites and trachybasalts), boninites, and high-Mg andesites (adakites) from arc settings[4]. This is because (1) alkaline rocks in arcs involve significant recycled components, which will result in $TiO_2$ enrichment; (2) boninites are formed by partial melting of mantle sources, which are relatively depleted due to their formation during

the early stage of subduction; and (3) high-Mg andesites (adakites) are derived from partial melting of oceanic slab and mixed with mantle peridotite. The outliers are discarded by using a modified Thompson-tau method. The detailed data filtering process is illustrated in the Supplementary Information. Accordingly, we collected a total of 2749 arc samples within 24 arc segments, 994 of which had V/Sc data and 833 of which had Cu/Zr data (Supplementary Data 2). For the primitive MORBs, we collected 759 samples, 442 of which had V/Sc data and 439

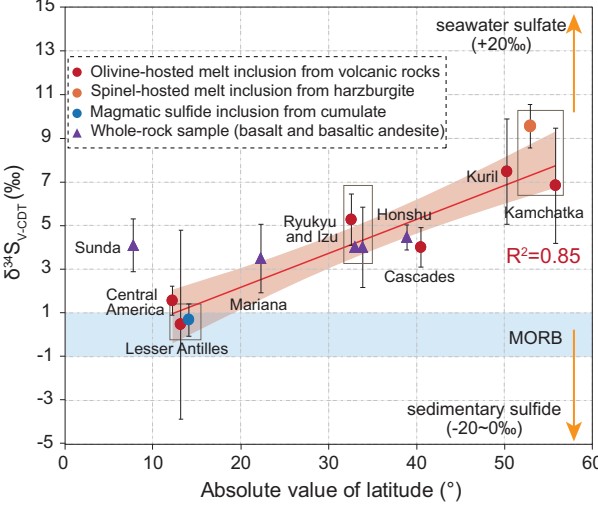

**Fig. 6 | Variation in average δ³⁴S (‰) of arc magma across latitude.** Symbols show average values and 1 SE (error bar) for each arc segment. Only melt inclusions with sulfur contents higher than 1000 ppm are selected. The whole-rock samples are selected based on their rock type, SiO₂ contents, or sulfur contents. δ³⁴S (‰) of MORB, seawater sulfate, and sedimentary sulfide are shown for comparison[80]. The shaded region of regression represents a 95% confidence interval. The compiled data are listed in Supplementary Data 15.

of which had Cu/Zr data (Supplementary Data 4). The arc samples are grouped based on their arc segments, whereas the MORBs are grouped based on latitude with a bin of 10°.

### Estimation of melting temperature and pressure
The estimation of melting temperature and pressure is based on the method of ref. 52 (Supplementary Data 2 and 4). As for the arc samples, the initial $Fe^{3+}/Fe_{Total}$ was set at 0.20[53], and the $H_2O$ content was set at 4.0 wt%[54]. As for the MORBs, the initial $Fe^{3+}/Fe_{Total}$ was set at 0.138[55], and the $H_2O$ content was set at 0.2 wt%[56]. The total compositions of both arc samples and MORBs are normalized to 100.0 wt%. During the corrections, the Fe–Mg exchange coefficient ($K_d$) between olivine and melt was set at 0.3[57].

### Modeling of redox state based on V/Sc–Ti system and Cu/Zr–Zr system
Near fractional melting was employed for both MORBs and arc basaltic rocks. In the modeling, the content of element $i$ in the instantaneous melt is given by $C_{i,m}^{n+1} = \frac{C_{i,r}^n}{D_{i,bulk}^{n+1} + F(1 - D_{i,bulk}^{n+1})}$, where $C_{i,m}^{n+1}$ is the content of element $i$ in the melt at step $n+1$, $C_{i,r}^n$ is the content of element $i$ in the residue after melt extraction at step $n$, $D_{i,bulk}^{n+1}$ is the bulk partition coefficient at step $n+1$, and $F$ is the melt fraction of each step. The element $i$ content in the aggregated melt at step $j$ is given by $C_i^{aggre} = \frac{\sum_1^j C_{i,m}^j M_{melt}^j}{\sum M_{melt}^j}$, where $C_{i,m}^j$ and $M_{melt}^j$ are the content of element $i$ in the melt and the mass of melt at step $j$, respectively. $C_i^{aggre}$ and $\sum M_{melt}^j$ are the content of element $i$ in the aggregated melt and bulk mass of the aggregated melt at step $j$, respectively.

The initial mantle mineral composition is considered as the spinel lherzolite, which is composed of 57% olivine (Ol), 28% orthopyroxene (Opx) ($Al^T = 0.15$, Al in the tetrahedron site; Wo# = 4.38 (Wo# = $X_{Wo}/(X_{Wo} + X_{En} + X_{Fs})$), where $X_{Wo}$, $X_{En}$, and $X_{Fs}$ are fractions of wollastonite, enstatite, and ferrosilite, respectively), 13% clinopyroxene (Cpx)

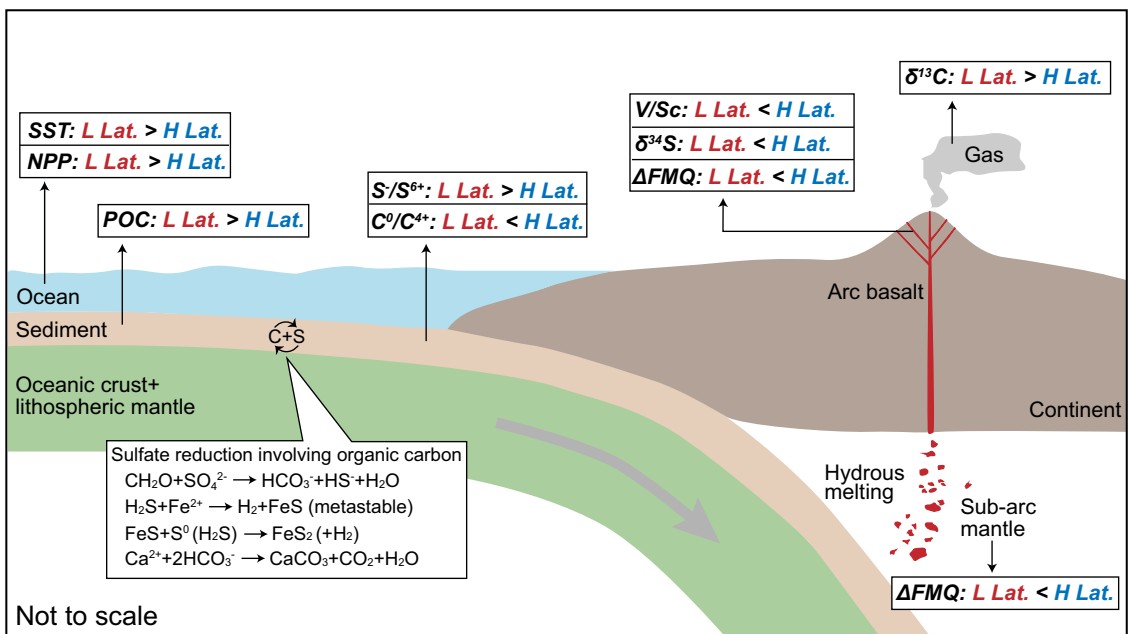

**Fig. 7 | Schematic diagram showing the influence of subducted sediments on the redox state of arc primary magma.** The different compositions of subducted sediments between the low latitudes and high latitudes result in different redox states of the sub-arc mantle. The high net primary production in the low-latitude regions leads to more particulate organic carbon deposited in the ocean floor. The more organic carbon could facilitate the sulfate reduction accompanied by the

formation of pyrite and carbonate[16]. The trench sediments in the low-latitude regions have higher ratios of sulfide/sulfate and carbonate/organic carbon, which will lead to a less oxidized sub-arc mantle compared to high-latitude arcs. The basalt in the low-latitude arcs exhibits lower V/Sc ratio and δ³⁴S values, and the volcanic gases have higher δ¹³C values. L low; H high; Lat. latitude; SST sea surface temperature; NPP net primary production; POC particulate organic carbon.

($Al^T = 0.17$), and 2% spinel (Sp) (Cr# = 10.7) based on ref. 58. The initial geochemical composition of depleted MOR mantle is V (79 ppm), Ti (798 ppm), Sc (16.3 ppm), Cu (30.0 ppm), and Zr (5.082 ppm)[58,59]. According to the compiled Sr–Nd isotopic compositions of these arc basaltic rocks (Fig. 3a), the initial geochemical composition of depleted arc mantle is calculated by 99% depleted MOR mantle plus 1% average subducted sediments[12], which is V (79.31 ppm), Ti (827.19 ppm), Sc (16.27 ppm), Cu (30.75 ppm), and Zr (6.331 ppm), respectively. The uncertainties of 7% for V, 12% for Ti, and 13% for Sc[59], and the range of Zr content from depleted (4.269 ppm) to enriched (6.087 ppm) MOR mantle[58] is also taken into consideration.

The bulk partition coefficient (bulk $D$-value) during mantle melting for an element $i$ at a given melting degree can be expressed as $D_i^{bulk} = \sum_k X^k \times D_i^{k/melt}$, where $X^k$ is the fraction of mineral $k$ in the residual mineral assemblage, and $D_i^{k/melt}$ is the partition coefficient for element $i$ between mineral $k$ and melt. The melting reactions from refs. 60,61 were used for anhydrous MORB mantle melting and hydrous arc mantle melting, respectively.

As for the V/Sc–Ti system, the recommended partition coefficients for different minerals are: $D_{Ti} = 0.008$ for Ol, $D_{Ti} = 0.30$ for Sp, $D_{Sc} = 0.12$ for Ol, $D_{Sc} = 0.06$ for Sp. $D_V$ for Ol, Opx, Cpx, and Sp and $D_{Sc}$ and $D_{Ti}$ for Opx and Cpx were calculated at the given pressure ($P$), temperature ($T$), and compositions of minerals and melt using equations from ref. 4. $P$ is the average melting pressure for arc basaltic rocks ($1.15 \pm 0.24$ GPa) or MORB ($1.20 \pm 0.20$ GPa). $T$ is the range of the calculated melting temperatures corresponding to their melting degree, with a melt productivity of 0.23/°C for lherzolite[60]. For arcs, the initial $T$ is 1200 °C, and the melting degree reaches 25% at 1295.2 °C (our calculated average melting temperature of arc basaltic rocks is $1231 \pm 33$ °C). For MORBs, the initial $T$ is 1300 °C, and the melting degree reaches 20% at 1387 °C (our calculated average melting temperature of MORB is $1342 \pm 29$ °C). The compositions of Opx, Cpx, and Sp are taken from ref. 58. The melt composition is the major element oxide of the sample, expressed as NBO/T and calculated according to equations from ref. 62. The calculated NBO/T values for arc samples and MORB are $1.12 \pm 0.14$ and $0.80 \pm 0.04$, respectively (Supplementary Data 2 and 4). The modeling results are listed in Supplementary Data 7 and 8. The potential uncertainty of this method is discussed in the Supplementary Information.

As for the Cu/Zr system, the Cu content in basaltic magma is controlled by oxygen fugacity and sulfur content in the mantle source. The Cu content in the melt is constrained by the bulk partition coefficient of Cu and the proportion of residual sulfide. The recommended partition coefficients for different minerals are $D_{Cu} = 0.047$ for Ol, $D_{Cu} = 0.0279$ for Opx, $D_{Cu} = 0.0601$ for Cpx, $D_{Cu} = 0.15$ for Sp, $D_{Cu} = 625$ or 850 for sulfide in arc mantle or MOR mantle, $D_{Zr} = 0.007$ for Ol, $D_{Zr} = 0.027$ for Opx, $D_{Zr} = 0.103$ for Cpx, and $D_{Zr} = 0.06$ for Sp[26,63–66]. The proportion of residual sulfide can be constrained by the sulfur content in the mantle, sulfur content in the mantle sulfide, and SCSS. The sulfur contents in the MOR mantle and arc mantle are different. According to previous studies, the sulfur content in MOR mantle is ~150–200 ppm[67,68], whereas the sulfur content in arc mantle is ~300–400 ppm[26,69]. The sulfur content in the mantle sulfide is ~36.9 wt % according to ref. 70. The SCSS can be calculated based on the redox state, melt composition, melting pressure, and temperature[18,71]. The calculated average SCSS for arc mantle and MOR mantle at $fO_2 = \Delta FMQ$ are $1463 \pm 229$ ppm and $1447 \pm 125$ ppm (Supplementary Data 2 and 4), respectively, based on the equations from ref. 71. The SCSS at $fO_2 > \Delta FMQ$ can be calculated according to the equations from ref. 18. The modeling results are listed in Supplementary Data 9 and 10.

### Variations of oxygen fugacity caused by redox budgets of C and S in subducted sediments

According to ref. 17, the input fluxes of $C^0$ and $C^{4+}$ are $(0.72 \pm 0.39) \times 10^{12}$ mol/year and $(1.18 \pm 0.60) \times 10^{12}$ mol/year,

respectively. In consideration of the uncertainty of the mantle reference state of carbon, we set the effective fluxes of $C^0$ and $C^{4+}$ to be half of their total fluxes. Therefore, the contributions to redox budget fluxes of $C^0$ and $C^{4+}$ are $-(1.44 \pm 0.78) \times 10^{12}$ mol/year and $(2.36 \pm 1.20) \times 10^{12}$ mol/year, respectively. According to refs. 45,46, the input flux of $S^{6+}$ is set to be a quarter or tenth of the flux of $S^-$, which is $(0.16 \pm 0.08) \times 10^{12}$ or $(0.62 \pm 0.33) \times 10^{11}$ mol/year. Therefore, the contributions to redox budget fluxes of $S^-$ and $S^{6+}$ are $(0.62 \pm 0.33) \times 10^{12}$ mol/year and $(1.28 \pm 0.64) \times 10^{12}$ or $(0.50 \pm 0.26) \times 10^{12}$ mol/year, respectively. Our primary goal is to evaluate the variations of oxygen fugacity caused by the variations of subducted sediments. Therefore, we considered the total variations of redox budgets as contribution fluxes of oxidant minus reductant, which means the redox budgets of carbon and sulfur in subducted sediments being $C^{4+} + S^- + S^{6+} - C^0$. Thus, the revised contribution fluxes of carbon and sulfur in subducted sediment should be varied between $2.75 \times 10^{12}$ and $8.65 \times 10^{12}$ mol/year or $2.34 \times 10^{12}$ and $7.49 \times 10^{12}$ mol/year. Such a range of redox budgets will result in the variations of $\Delta FMQ$ between 0.3 and 1.2 log units at a time scale of 1–10 million years.

### Data availability

All geochemical and relevant data used in this study are available at Figshare (https://doi.org/10.6084/m9.figshare.24551521). Source data are provided with this paper.

### Code availability

The code used for generating the map of Fig. 1 and plotting data on map is available at Figshare (https://doi.org/10.6084/m9.figshare.24551521).

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

## Acknowledgements

F.Y.H., H.H.J., B.W., and F.Y.W. acknowledge the financial support from the NSFC (92255303, 42325206, 42488201), the Strategy Priority Research Program (Category B) of the Chinese Academy of Sciences (No. XDB0710000). M.N.D. was funded by the program (PNRR-III-C9-2023–I8) of Romanian MCID (C1.2.PFE-CDI.2021-587/Contract No. 41PFE/30.12.2021 and 64/30.07.2023).

## Author contributions

B.W. and F.Y.W. conceived the study. F.Y.H. designed the study and completed the data selection and filtration. F.Y.H. and L.G. carried out the geochemical modeling of variations of V/Sc and Cu/Zr during mantle melting at different oxygen fugacity conditions. F.Y.H., H.H.J., B.W., and M.N.D. led the writing of the paper with contributions from all authors.

## Competing interests

The authors declare no competing interests.
