## [Peer Review File · Nature Communications]

REVIEWERS' COMMENTS

Reviewer #1 (Remarks to the Author):

I reviewed the previous version of this paper submitted to Nature Geoscience. As I mentioned in an earlier review that their observations on latitude-dependent oxygen fugacity in arc magmas are important discoveries and publication of this work would inspire future researchers to investigate the mechanism responsible for such observations after revision.

In this version, the authors have diligently addressed the previous comments and revised their manuscript accordingly. They have proposed a new mechanism emphasizing the combined effect of S and C in regulating the redox state of arc volcanics for the observed redox variations, which is more plausible compared to the earlier model of serpentinite-induced carbonate reduction. They also provided sulfur isotope in global arc volcanics as new evidence supporting it. They also recalculated the redox budget accordingly, not just based on the C budget as calculated in the previous version. The authors have also candidly acknowledged the current limitations of their study. However, these limitations do not undermine the significance of their findings. I anticipate that the scientific community will swiftly respond to this work. The publication of this study is likely to stimulate future research aimed at elucidating the mechanism responsible for observed redox variations globally.

Specific comments

1. Line 344: “hare” should be “are”
2. There are 15 supplementary tables in this work and it'll be better to provide the table captions in the supplementary document or somewhere else. This way, the readers would easily find the information that they are interested in.
3. It'll be clearer if the authors could add more details for the redox calculation part in the method section.

Response to comments

Reviewer #1 (Remarks to the Author):

I reviewed the previous version of this paper submitted to Nature Geoscience. As I mentioned in an earlier review that their observations on latitude-dependent oxygen fugacity in arc magmas are important discoveries and publication of this work would inspire future researchers to investigate the mechanism responsible for such observations after revision.

In this version, the authors have diligently addressed the previous comments and revised their manuscript accordingly. They have proposed a new mechanism emphasizing the combined effect of S and C in regulating the redox state of arc volcanics for the observed redox variations, which is more plausible compared to the earlier model of serpentinite-induced carbonate reduction. They also provided sulfur isotope in global arc volcanics as new evidence supporting it. They also recalculated the redox budget accordingly, not just based on the C budget as calculated in the previous version. The authors have also candidly acknowledged the current limitations of their study. However, these limitations do not undermine the significance of their findings. I anticipate that the scientific community will swiftly respond to this work. The publication of this study is likely to stimulate future research aimed at elucidating the mechanism responsible for observed redox variations globally.

Specific comments

1. Line 344: “hare” should be “are”.

Revised. We have changed ‘hare’ to ‘are’. Please see new Line 264.

2. There are 15 supplementary tables in this work and it'll be better to provide the table captions in the supplementary document or somewhere else. This way, the readers would easily find the information that they are interested in.

Revised. We have added a txt file listing all the table captions of supplementary tables deposited in the zipped file as Supplementary Dataset.

3. It'll be clearer if the authors could add more details for the redox calculation part in the method section.

Revised. We have added more details for redox calculations. Please see the revised Methods.

‘According to Evans (2010), the input fluxes of C^0 and C^{4+} are $(0.72 \pm 0.39) \times 10^{12}$ mol/year and $(1.18 \pm 0.60) \times 10^{12}$ mol/year, respectively. In consideration of the uncertainty of the mantle reference state of carbon, we set the effective fluxes of C^0 and C^{4+} to be half of their total fluxes. Therefore, the contributions to redox budget fluxes of C^0 and C^{4+} are $-(1.44 \pm 0.78) \times 10^{12}$ mol/year and $(2.36 \pm 1.20) \times 10^{12}$ mol/year, respectively. According to Brounce et al. (2019) and de Moor et al. (2022), the input flux of S^{6+} is set to be a quarter or tenth of the flux of S^- , which is $(0.16 \pm 0.08) \times 10^{12}$ or $(0.62 \pm 0.33) \times 10^{11}$ mol/year. Therefore, the contributions to redox budget fluxes of S^- and S^{6+} are $(0.62 \pm 0.33) \times 10^{12}$ mol/year and $(1.28 \pm 0.64) \times 10^{12}$ or $(0.50 \pm 0.26) \times 10^{12}$

mol/year, respectively. Our primary goal is to evaluate the variations of oxygen fugacity caused by the variations of subducted sediments. Therefore, we consider the total variations of redox budgets as contribution fluxes of oxidant minus reductant, which means the redox budgets of carbon and sulfur in subducted sediments being $C^{4+}+S^{-}+S^{6+}-C^0$. Thus, the revised contribution fluxes of carbon and sulfur in subducted sediment should be varied between 2.75×10^{12} and 8.65×10^{12} mol/year or 2.34×10^{12} and 7.49×10^{12} mol/year. Such a range of redox budgets will result in the variations of ΔFMQ between 0.3 to 1.2 log units at a time scale of 1 to 10 million years.'